# Metal-hydrogen systems with an exceptionally large and tunable thermodynamic destabilization

Peter Ngene[1], Alessandro Longo[2,3], Lennard Mooij[4], Wim Bras[3] & Bernard Dam [4]

Hydrogen is a key element in the energy transition. Hydrogen–metal systems have been studied for various energy-related applications, e.g., for their use in reversible hydrogen storage, catalysis, hydrogen sensing, and rechargeable batteries. These applications depend strongly on the thermodynamics of the metal–hydrogen system. Therefore, tailoring the thermodynamics of metal–hydrogen interactions is crucial for tuning the properties of metal hydrides. Here we present a case of large metal hydride destabilization by elastic strain. The addition of small amounts of zirconium to yttrium leads to a compression of the yttrium lattice, which is maintained during (de)hydrogenation cycles. As a result, the equilibrium hydrogen pressure of $YH_2 \leftrightarrow YH_3$ can be rationally and precisely tuned up to five orders of magnitude at room temperature. This allows us to realize a hydrogen sensor which indicates the ambient hydrogen pressure over four orders of magnitude by an eye-visible color change.

[1] Inorganic Chemistry and Catalysis, Debye Institute for Nanomaterials Science, Utrecht University, Universiteitsweg 99, Utrecht, 3584 CG, The Netherlands. [2] Istituto per lo Studio dei Materiali Nanostrutturati ISMN-CNR, Palermo, Via Ugo La Malfa, 153, 90146 Palermo, Italy. [3] Netherlands Organization for Scientific Research (NWO), Dutch-Belgian Beamline, ESRF—The European Synchrotron, CS40220, 38043, 71 Avenue des Martyrs, 38000 Grenoble, France. [4] Materials for Energy Conversion and Storage (MECS), Department of Chemical Engineering, Delft University of Technology, Van der Maasweg 9, Delft, 2629 HZ, The Netherlands. Correspondence and requests for materials should be addressed to P.N. (email: p.ngene@uu.nl)

Reversible hydrogen absorption in metals is exploited for a variety of applications, such as hydrogen storage[1, 2], hydrogen sensing[3], rechargeable batteries[2, 4, 5], smart windows[3, 6], hydrogen purification/separation[7, 8], and catalysis[9]. For most applications, it is essential to tune the temperature and pressure at which metals can reversibly absorb hydrogen. This is essentially governed by the thermodynamics of the metal–hydrogen interaction[1, 10]. Hence, over the past three decades, much effort has been made to tune the thermodynamics of metal–hydrogen interactions[10–20]. The equilibrium temperature and pressure at which a metal can absorb and desorb hydrogen is a fundamental property of the metal which is simply described by the Van't Hoff relation (Eq. (1)), when neglecting the temperature dependence of the enthalpy and entropy).

$$\ln\left(\frac{P_{eq}}{P_0}\right) = \frac{\Delta H^0}{RT} - \frac{\Delta S^0}{R} \qquad (1)$$

Here, $P_{eq}$ is the equilibrium or plateau hydrogen pressure, $P_o$ is the standard pressure (taken as 1 bar $H_2$), $R$ is the gas constant, $T$ is the temperature, $\Delta H^0$ and $\Delta S^0$ are the change in enthalpy and entropy, respectively, at standard conditions, accompanying the (de)hydrogenation reaction. The entropy change associated with the transition of hydrogen in the gas phase into chemisorbed hydrogen in the metal hydride phase is similar for most metal–hydrogen systems ($130\,J\,K^{-1}\,mol_{H_2}^{-1}$). Therefore, the thermodynamic properties of metal–hydrogen systems are usually evaluated by the enthalpy of the (de)hydrogenation reaction, assuming that entropy changes in the lattice are negligible. In the absence of any kinetic limitation, an increase or decrease of the plateau hydrogenation pressure ($P_{eq}$) implies that the metal hydride phase is destabilized or stabilized, respectively. For hydrogen storage applications, an equilibrium pressure of 1–10 bar at temperature between 25 and 150 °C is required. Therefore, destabilization is crucial for the use of stable metal hydrides such as LiH, $MgH_2$, and $YH_3$ (low-equilibrium pressure or high-dehydrogenation temperatures), while stabilization is necessary for unstable/metastable hydrides such as $AlH_3$ and $LiAlH_4$. Similar tuning is required for battery and sensor applications. Strategies such as alloying, nano-sizing, nano-confinement, and interfacial effects have been tested to (de)stabilize or tune the thermodynamics of metal–hydrogen systems[11–20]. However, none of these approaches has so far led to a controllable and large enough tuning of the equilibrium pressure of hydrogenation.

Here we report on a metal–hydrogen system that exhibits a tunable equilibrium hydrogen pressure over more than five orders of magnitude at room temperature. The addition of 0.1–13 atomic% Zr to Y leads to a continuous and well-defined increase in the equilibrium hydrogenation pressure for the formation of yttrium trihydride ($YH_3$), from $10^{-1}$ to $10^4$ mbar at room temperature. Also, the dehydrogenation pressure increases, although to a lesser extent. Structural characterization suggests that this remarkable thermodynamic destabilization is due to the physical constraints (lattice strain) arising from a compression of the $YH_x$ matrix induced by $ZrH_x$ nanoclusters. The effect is maintained on cycling, and is accompanied by a strong optical change due to the $YH_2 \rightarrow YH_3$ metal insulator transition. At this transition, metallic face centered cubic (fcc) $YH_2$ transforms into the hexagonal $YH_3$ phase with a bandgap of around 2.7 eV[21]. The metallic fcc phase has a solubility range, in which the optical reflection increases with the hydrogen content. On hydrogenation, the characteristic transmittance window in the range of 1.5–2.0 eV observed at $YH_{1.9}$, closes at $YH_{2.1}$[6, 21, 22]. These optical changes allow the design of eye-readable hydrogen detectors discriminating between four distinct pressure levels[23]. Based on the ability to precisely tune the pressure level corresponding to the $YH_2 \rightarrow YH_3$ phase transition, here we demonstrate an optical hydrogen sensor with a detection range spanning more than four orders in magnitude at room temperature.

## Results

**The effect of Zr on the equilibrium pressure of Y–H system.** $Y_xZr_{1-x}$ thin films were deposited on quartz substrates by co-sputtering in a RF-DC magnetron sputter system. All films are covered with Pd to prevent oxidation and to catalyze hydrogen sorption. In addition, a 5 nm Ti layer is deposited between the Y and Pd layers to prevent alloying. Gradient $Y_xZr_{1-x}$ thin films were deposited on a $70 \times 5$ mm quartz substrate in such a way that the concentration of Y and Zr varies almost linearly along the substrate, as shown in Fig. 1a. Due to the different molar volumes, the film thickness reduces by more than 50% along the sample length. Hydrogen sorption was monitored by an optical technique called hydrogenography[24]. This method allows one to measure pressure-optical transmission isotherms (PTIs), which provide the same information as the typical pressure-composition-isotherms (PCIs) but with the advantage that many samples can be measured simultaneously at exactly the same conditions.

In Fig. 1b, we show the PTIs for 73 distinct points (Zr concentrations) along the sample length, starting at $Y_{98.5}Zr_{1.5}$ and ending at $Y_{93.25}Zr_{6.75}$. Three different optical transitions are observed during this first hydrogenation cycle. These are due to the optical changes accompanying the transitions $Y \rightarrow YH_{1.9} \rightarrow YH_{2.1} \rightarrow YH_3$, respectively[23]. The transition from metallic Y to semitransparent $YH_{1.9}$ starts at around $5.10^{-3}$ mbar and is completed at around 0.15 mbar. It is immediately followed by the formation of $YH_{2.1}$, which is slightly darker than $YH_{1.9}$, leading to a small decrease in transmittance. The reported equilibrium pressure for the $Y \rightarrow YH_{1.9}$ transition is $\sim 10^{-31}$ mbar[25, 26], which is much lower than what we observe here. This indicates that this phase transition is kinetically limited, as suggested by the tilted nature of the isotherm ($\sim 5 \times 10^{-3}$–0.15 mbar). We also observed a similar (tilted) isotherm in a Zr-free reference sample with similar gradient in Y thickness (Supplementary Fig. 1). In contrast, the well-defined plateau pressure and flat nature of the $YH_{2.1} \rightarrow YH_3$ shows that this transition does not suffer from kinetic limitations as in $Y \rightarrow YH_{1.9}$. Moreover, the 0.1 mbar plateau pressure of the reference sample (Supplementary Fig. 1) fits the thermodynamic data for pure Y[25]. Remarkably, the $YH_{2.1} \rightarrow YH_3$ plateau pressure shows a profound dependence on the Zr concentration, increasing continuously from 0.3 mbar for $Y_{98.5}Zr_{1.5}$ to 40 mbar for $Y_{93.25}Zr_{6.75}$ (Fig. 1b). Clearly, the presence of Zr has no noticeable effect on the kinetically driven $Y \rightarrow YH_{2.1}$ phase transitions. The decrease in width of the isotherms with increasing Zr concentration is due to a decrease in film thickness, leading to a decrease in the amount of hydrogen absorbed and hence the magnitude of the optical change. Note, also that the change in optical transmittance of Zr upon hydrogenation is negligible compared to that of Y (Supplementary Fig. 2). $YH_3$ cannot be fully dehydrogenated to Y at moderate conditions; instead, the sample remains in the $YH_{1.9}$ phase. Therefore, after the first hydrogenation/dehydrogenation cycle is completed, only the $YH_{1.9} \rightarrow YH_{2.1} \rightarrow YH_3$ transitions are observed in subsequent cycles.

Figure 1c shows the isotherms (at 25 °C) for the sixth hydrogenation cycle. All isotherms show two distinct plateau pressures, one corresponding to $YH_{2.1} \rightarrow YH_3$ and the other around 40–45 mbar due to hydrogenation of the Pd layer. The slight decrease in transmittance in $\ln(T/T_0)$ from 0 to −0.3 at low pressures is again due to the hydrogenation within the solid solution ($YH_{1.9} \rightarrow YH_{2.1}$). As expected, the equilibrium hydrogen pressure for $Pd \rightarrow PdH_x$ does not change with the Zr

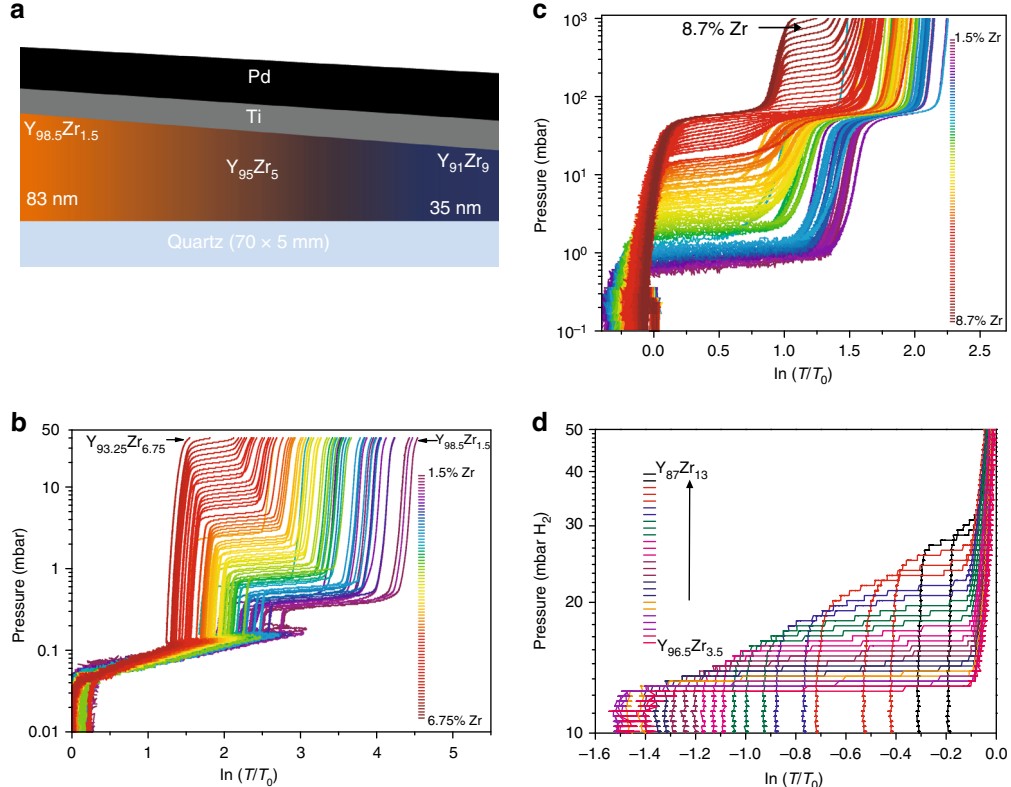

**Fig. 1** The effect of Zr on the hydrogenation pressure of Y. **a** Schematic picture of the Y–Zr compositional gradient resulting in a wedge geometry with a thickness ranging from ~38 to 80 nm on a 70 × 5 cm quartz substrate, topped by a 5 nm Ti and 30 nm Pd cap layer. **b** Pressure transmission isotherms (PTIs) during the first hydrogen absorption cycle (0–40 mbar $H_2$ at 25 °C). It shows the PTIs of 73 different Zr concentrations measured simultaneously in the thin film compositional gradient. Clearly, the pressure at which the $YH_{2.1} \rightarrow YH_3$ transition occurs depends on the Zr concentration. Each line in the graph represents a 0.015 at.% increase in Zr, starting from 1.5% Zr, i.e., $Y_{98.5}Zr_{1.5}$. **c** PTIs of the sample during the sixth hydrogen absorption cycle at 25 °C, starting from the dihydride state. **d** Desorption isotherm (PTIs) at 220 °C for selected Zr concentrations ($Y_{96.6}Zr_{3.4}$–$Y_{85.5}Zr_{14.5}$). Each isotherm represents a 0.38 at. % increase in Zr, starting from 3.5% Zr. Note that before dehydrogenation, the sample was first exposed to $10^4$ mbar (10 bar) $H_2$. The steps in pressure are due to due to the logarithmic decrease in hydrogen pressure from $10^4$ to 1 mbar

concentration, while that of $YH_{2.1} \rightarrow YH_3$ increases from ~ 0.3 mbar for 1.5% Zr to 1000 mbar for ~8.7% Zr ($Y_{91.3}Zr_{8.7}$). At about 7% Zr, the plateau pressure for the $YH_{2.1} \rightarrow YH_3$ transition is close to that of $Pd \rightarrow PdH_x$, hence the sample seems to have one long plateau pressure around this composition. Above 7% Zr, the plateau pressure for $YH_{2.1} \rightarrow YH_3$ is clearly higher than that of $Pd \rightarrow PdH$, resulting in two plateaus again. In Supplementary Fig. 3, we show that Y containing about 13 atom % Zr requires up to $10^4$ mbar for full hydrogenation to $YH_3$. However, at these high Zr concentrations, the isotherms are tilted, thereby complicating the determination of the equilibrium pressure. Possibly, part of the tilt evolves because the hydrogen pressure is increased logarithmically from 1 to $10^4$ mbar, resulting in a higher step size at higher pressures.

To exclude any effects of the thickness gradient (Supplementary Fig. 1) and/or lateral interaction between the various compositions, we repeated the hydrogenation experiments on six separate samples (1 × 1 cm) having the same total thickness (60 nm Y–Zr layer), but different Zr concentrations (Supplementary Fig. 4). Again, we observe that the equilibrium hydrogen pressure increases with the Zr concentration. For ~13 at.% Zr, the hydrogen plateau pressure approaches $10^4$ mbar (the limit of our equipment) as shown in Supplementary Figs. 4 and 5. Thus, the addition of only 13 at.% Zr to Y leads to about five orders of magnitude increase in plateau pressure for the formation of $YH_3$. This indicates an exponential relationship between the hydrogen plateau pressure and the Zr concentration as depicted in

Supplementary Fig. 5. To our knowledge, such a strong effect of doping on the plateau pressure has never been observed before.

True thermodynamic destabilization should also lead to an increase in the equilibrium dehydrogenation pressure. Indeed, we observe this to be the case, albeit to a lesser extent (Fig. 1d). Yttrium thin films are known to exhibit a large hysteresis between the equilibrium hydrogenation and dehydrogenation pressure. This is due to first-order nature of the transition[27]. Consequently, dehydrogenation experiments were conducted between 200 and 250 °C. At these temperatures, we find that the equilibrium dehydrogenation pressure $Y_{87}Zr_{13}H_x$ is about three times higher than for pure $YH_3$ (~30 vs. 10.5 mbar (Supplementary Fig. 1b). The asymmetry in destabilization for hydrogenation and dehydrogenation might be related to the different structural changes that occur during these processes[28]. Without going in detail here, we find that the GdZr–H system exhibits similar tunable plateau pressures (Supplementary Fig. 6), showing that this effect is not limited to Y but probably extends to all the rare earth hydrides having a similar hydrogenation behavior.

**The effect of Zr on the Y lattice**. To understand the origin of this remarkable thermodynamic effect, we used in situ XRD measurements to gain insight into the structure of the samples at different stages; as-prepared, hydrogenated, and dehydrogenated. The as-prepared/sputtered Y is hexagonal close packed (hcp) and exhibits a preferential orientation with the Y(002) direction

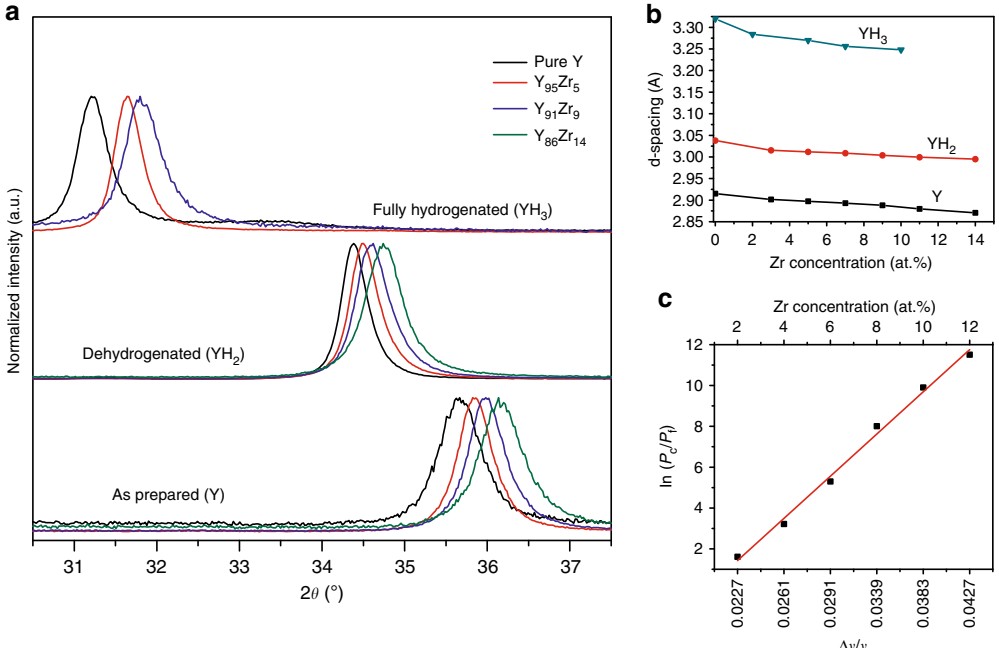

**Fig. 2** Lattice behavior of YH$_x$ as a function of the Zr doping. **a** XRD pattern of Y–Zr thin films showing the hcp-Y(002) diffraction peak in the as-prepared, fully hydrogenated hcp-YH$_3$ (002) and dehydrogenated YH$_2$(111). The position of the peak clearly depends on the Zr concentration. Note that the Y$_{86}$Zr$_{14}$ sample could not be fully hydrogenated to YH$_3$ even when applying 10 bar H$_2$ for 24 h. **b** Change in the corresponding YH$_x$ d-spacing as a function of the Zr concentration in the as-prepared, fully hydrogenated (YH$_3$), and dehydrogenated (YH$_2$) state. **c** Relation between the relative change in molar volume ($\Delta V/V$) of YH$_2$ and the plateau pressure at the YH$_2 \leftrightarrow$ YH$_3$ phase transition comparing uncompressed ($P_f$) and compressed ($P_C$) YH$_2$

perpendicular to the substrate (Fig. 2a). Upon hydrogenation, it transforms into fcc YH$_{1.9–2.1}$ and subsequently into hcp YH$_3$. The optical transition accompanying YH$_{1.9} \rightarrow$ YH$_{2.1}$ does not result from a structural change but from an intra-band transition within the fcc phase[26, 29]. The YH$_{1.9–2.1}$ solid solution phase is therefore simply denoted as YH$_2$. No sign of crystalline Zr is seen in the Y$_x$Zr$_{1-x}$ thin films after preparation, hydrogenation, or dehydrogenation. However, the $2\theta$ value of Y(002) diffraction peak increases substantially in the Y$_x$Zr$_{1-x}$ films compared to that of pure hcp Y, suggesting the formation of an Y–Zr alloy. Although Y and Zr are immiscible[30], the high energy of the sputtering process might have enabled the formation of a metastable Y–Zr alloy or a structurally coherent Y–Zr system, as previously reported for some other alloys with a positive heat of mixing[31–33]. The increase in the Y(002) peak position implies a decrease in the inter-planar distances (d-spacing) of Y and thus suggests a compression of Y lattice due to the smaller atomic radius of Zr. Indeed, the lattice compression increases with the Zr concentration. A similar compression is also observed in the YH$_2$ and YH$_3$ phases (Fig. 2a, b, Supplementary Fig. 7, and Supplementary Table 1) showing that the lattice compression is maintained during the hydrogen absorption and desorption processes.

It is well known that the dissolution of hydrogen atoms in some metals leads to an expansion of the metal lattice by 2–3 Å$^3$ per hydrogen atom. This lattice dilatation favors absorption of additional H atoms, resulting in a net H–H attraction. However, if the metal lattice is constrained from expansion, this interaction may weaken and thereby increase the enthalpy of (de)hydrogenation. Consequently, the enthalpy of hydride formation ($\Delta H$) is related to the volume expansion according to Eq. (2)[34].

$$\frac{d\Delta H}{d\ln V} = -BV_H \qquad (2)$$

Here, $B$ is the bulk modulus of the metal and $V_H$ is the partial molar volume of hydrogen in the metal hydride. The equation

implies that a relative change in the molar volume ($\Delta V/V$) of the metal leads to a change in the plateau pressure according to Eq. (3), provided that the volume compression is elastic, hence remains intact upon hydrogenation of the metal.

$$\ln\left(\frac{P_c}{P_f}\right) = 2\frac{BV_H}{RT}\frac{\Delta V}{V} \qquad (3)$$

$P_c$ and $P_f$ are the plateau pressure of the compressed and uncompressed system, respectively, $R$ is the gas constant and $T$ the temperature. This equation implies that the thermodynamics of hydrogen sorption in metal can be precisely tuned by elastic clamping[35]. In Fig. 2c, $\ln(P_c/P_f)$ is plotted against the relative change in molar volume ($\Delta V/V$) of the YH$_2$ as a function of the Zr concentration. This plot gives a linear relationship, as expected from Eq. (3). For Y$_{87}$Zr$_{13}$, $P_c/P_f$ is ~10$^5$ at room temperature (Supplementary Figs. 4 and 5), which implies that a 5.0% volume compression ($\Delta V/V$) is expected for YH$_2$ based on Eq. (3), using 140 GPa as the bulk modulus $B$ of YH$_{2+x}$[36, 37] and $2 \times 10^{-6}$ m$^3$ mol$^{-1}$ for the partial molar volume of hydrogen. This is very close to 4.3% compression observed for the YH$_2$ phase in this sample (Fig. 2b and and Supplementary Table 1), which suggests that elastic compression is responsible for the thermodynamic effects observed upon Zr doping. As mentioned earlier, the Y $\rightarrow$ YH$_2$ transition is dominated by kinetics which hinders the observation of such thermodynamic effects.

**The structure of Zr within the yttrium matrix.** It is intriguing to observe that this large compression of the YH$_x$ lattice remains intact during the hydrogenation/dehydrogenation process, which involves significant structural transformations and plastic deformations. Therefore, we investigated the structure and local environment of the ZrH$_x$ and YH$_x$ phases using in situ XAS (X-ray absorption spectroscopy). Figure 3 shows the X-ray absorption near edge spectra (XANES) and extended X-ray absorption fine structures (EXAFS) of the Zr K-edge of the samples at

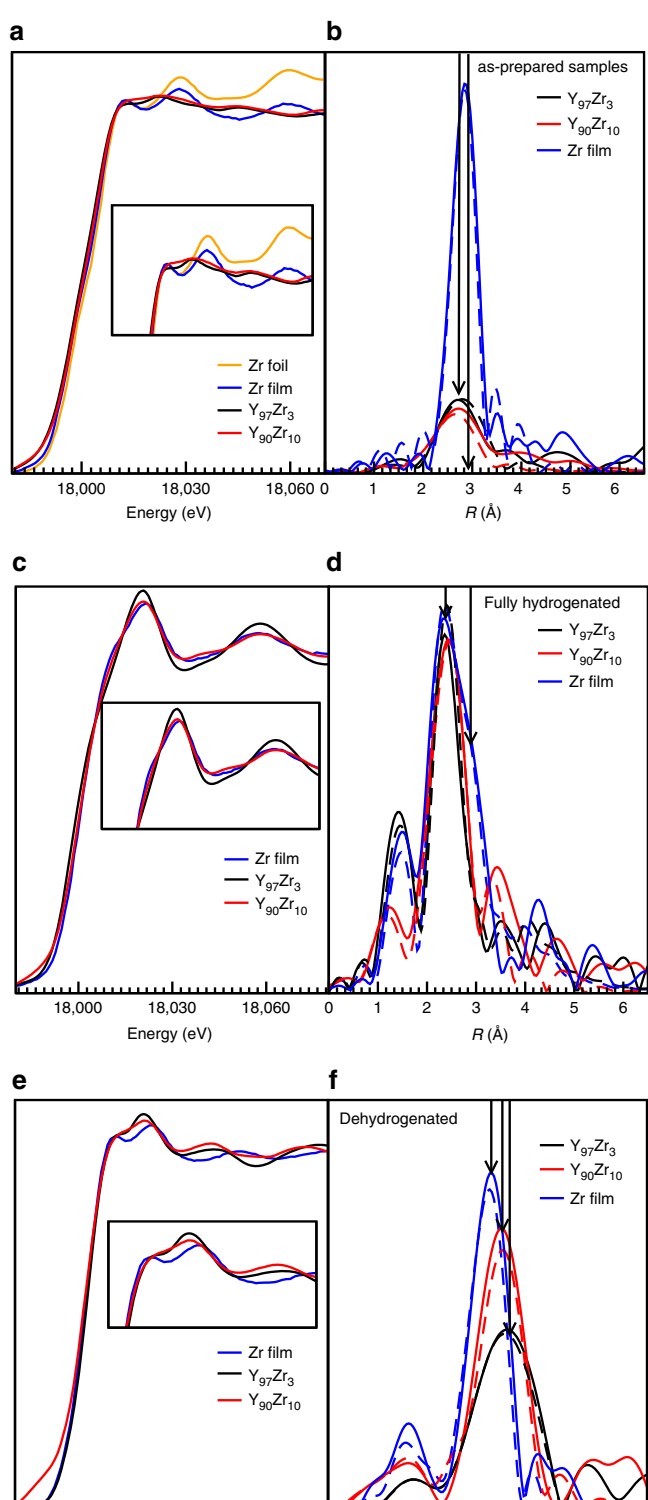

**Fig. 3** Local structure analysis with XAS. XANES spectra of (**a**) the as-deposited, (**c**) fully hydrogenated, and (**e**) dehydrogenated thin films. Phase uncorrected FT of the EXAFS spectra of the (**b**) as-deposited, (**d**) fully hydrogenated, and (**f**) dehydrogenated Zr and Y–Zr thin films. The dashed lines are the fits of the EXAFS data, and the arrows are guides to the eye

different stages. As shown in Fig. 3a, the as-sputtered Zr thin film (50 nm) has spectra that are similar to that of a standard Zr reference foil, but with substantially reduced amplitudes. This suggests that the sputtered Zr film has a similar atomic arrangement as bulk Zr, but with reduced size features as expected for a 50 nm film. The difference between the Zr and YZr films is evident in Fig. 3b, which shows the Fourier transform (FT) of their EXAFS spectra. Only one first neighbor peak is seen in the YZr samples, and the amplitude of the peak is significantly lower than for the sputtered Zr thin film. This points to a reduction in the Zr–Zr coordination number in the YZr samples compared to the Zr film (Supplementary Table 2 and Supplementary Fig. 8) and suggests that in the as-deposited YZr thin films, Zr forms nanoclusters[17] rather than being atomically dissolved in the Y matrix. If Zr were atomically dissolved in yttrium, Zr would see the environment of the Y cluster which has a large coherence length according to the XRD, and therefore it would have a high coordination number. Therefore, the fact that the Y lattice is compressed is not due to the formation of a dispersed Y–Zr alloy. Instead, we postulate the formation of a coherent hcp YZr system[38–40] consisting of Zr clusters coherently coupled to an yttrium matrix. The fact that we do not observe a slightly expanded first shell Zr–Zr coordination distance is probably due to the small size of the clusters.

Comparing Fig. 3a, b to Fig. 3c, d reveals that hydrogenation (1.5 bar $H_2$ at 25 °C) induces profound changes in the structure and local environment of Zr. Based on the enthalpy of formation, exposure of the sample to hydrogen should result first in the formation of $YH_2$ and $ZrH_x$, followed by the formation of $YH_3$ at elevated hydrogen pressures[41, 42]. For pure Zr-hydride thin film, the analysis suggests two contributions: one from a cubic ($\delta$–$ZrH_{1.4-1.7}$) and the other from a tetragonal ($\epsilon$–$ZrH_{1.7-2}$) zirconium dihydride phase. In YZr hydride, both the Zr coordination and the Zr–Zr distance are reduced as compared to pure Zr hydride. The latter is strange, since we would expect an expanded lattice if the assumption of a coherent lattice is correct.

To resolve this issue, we looked at the XANES spectra in more detail. First, we simulated the spectrum of the hydrogenated Zr film ($ZrH_x$) on the basis of the EXAFS data (details in Supplementary Methods section). As shown in Fig. 4a, we find that a phase mixture of 30% cubic and 70% tetragonal clusters with a 5 Å radius closely matches the observed spectrum. As shown in Supplementary Figs. 9–17, such a match cannot be obtained in a single phase sample, even if we assume very small clusters. For the YZr hydride samples, it proved to be impossible to simulate the XANES spectrum with either the cubic or the tetragonal phase or a mixture thereof (Fig. 4b). Instead, we assumed a hexagonal Zr packing with lattice parameters ($a = b = 2.90$ Å, $c = 5.90$ Å), which are similar to those derived from the Y K-edge EXAFS results of the fully hydrogenated YZr film (see Supplementary Tables 3 and 4 and Supplementary Figs. 18–20 for the XANES and EXAFS results of the Y K-edge). The 7 Å cluster was then computationally relaxed to the lowest energy minimum and the XANES spectrum was calculated for the now strongly distorted cluster. To our surprise, a reasonable agreement of the measured and calculated spectrum is thus obtained (Fig. 4c). The fact that the lattice symmetry matches that of $YH_3$ is a strong indication for a coherent coupling of the two lattices in the hydrogenated state. Thus, we conclude that $ZrH_x$ is forced to adopt the hexagonal symmetry of the yttrium trihydride to maintain the coherent interface at the fully hydrogenated state. Such behavior has been observed in perovskite oxide films, where for example, $La_{0.67}Sr_{0.33}MnO_3$ films grown on $SrTiO_3$ substrate assumes the symmetry of the cubic $SrTiO_3$ at the $La_{0.67}Sr_{0.33}MnO_3/SrTiO_3$ interface, leading to suppression of the octahedral rotations and an elongated c-axis lattice parameter[43].

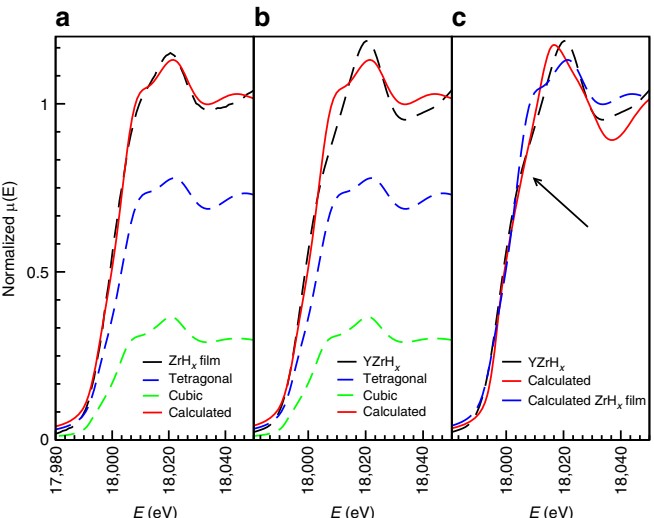

**Fig. 4** XANES simulations performed on hydrogenated Zr and YZr thin films. **a** ZrH$_x$ film, **b**, **c** YZrH$_x$ thin films. The arrow in **c** highlights the different position of the shoulder in the YZrH$_x$ sample, which cannot be reproduced by using either tetragonal or cubic phase

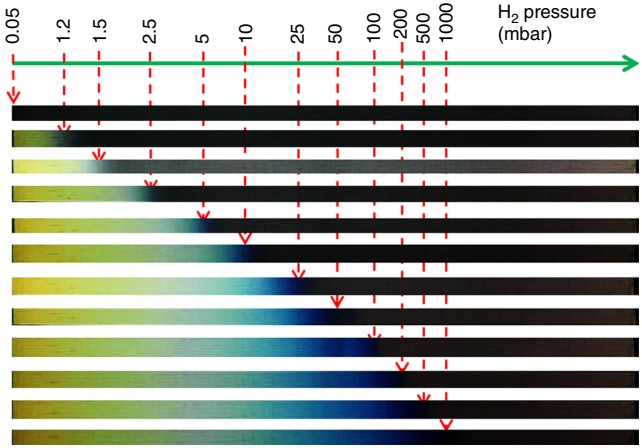

**Fig. 5** Visible color changes associated with the formation of YH$_3$ in a 70 × 5 mm Y$_{97}$Zr$_3$–Y$_{90}$Zr$_{10}$ gradient thin film. The pictures represent consecutive stages in time of the thin film hydrogenation at different H$_2$ pressures. The positions of the color front (indicated by the arrows) clearly depend on the hydrogen pressure and Zr concentration. Note that the thickness of the Pd cap layer used for this sample is 50 nm so as to enhance the optical contrast of the film. The sample is viewed from the substrate side

Upon dehydrogenation, the EXAFS data show the presence of a cubic (δ–ZrH$_x$) phase in all samples. Interestingly, the Zr–Zr interatomic distances in the YZr samples are significantly larger than in the pure Zr sample (6 Å vs. 4 Å). This suggests that the fcc ZrH$_x$ lattice is expanded in order to maintain a coherent interface with the slightly larger lattice of the fcc YH$_2$ phase. Upon subsequent (de)hydrogenation cycles, the samples exhibit XANES and EXAFS spectra that are similar to Fig. 3c–f, respectively, showing that the lattice compression and microstructure of the samples are stable upon cycling. Note that the transformation from cubic to hexagonal ZrH$_x$ during hydrogenation, and vise versa during dehydrogenation, leads to a ~12–17.2% volume change[44]. The volume expansion and contraction will lead to a significant asymmetry in the hydrogenation and dehydrogenation processes. This might be the reason for the observed differences in the plateau pressures of the hydrogenation and dehydrogenation processes. Note, also that the Y K-edge XAS results agrees with the XRD results which shows decreasing d-spacing of YH$_x$ lattice with increasing Zr concentration (see Supplementary Tables 3 and 4 and Supplementary Figs. 18–20 for the results of the Y K-edge XAS).

The results from the XRD and XAS measurements strongly indicate that the remarkable thermodynamic effect observed in this system is due to lattice compression of YH$_x$ by ZH$_x$ nanoclusters in a coherently coupled YH$_x$–ZrH$_x$ system. The driving force for such a coherent interface is the small sizes of the Zr (clusters) for which the interface energy is reduced by coherency at the expense of strain (compression) energy[45]. It has been shown that for small particles, the strain energy due to coherency can be more than compensated by the reduction in surface energy from interfacial chemical bonding[45, 46]. The large compression of the Y lattice by small amounts of Zr (4.3% compression by ~12.5 at.% Zr) is likely due to the low bulk modulus of yttrium metal (41.2 GPa, compared to 91 GPa for Zr). The compressive stress required to achieve this lattice dilation is only 1.77 GPa. However, the bulk modulus of Y increases sharply upon hydrogen absorption (up to 140 GPa for YH$_{2+x}$)[36, 37] thereby increasing the compressive stress to 6.02 GPa. It is also interesting that YH$_x$ can apparently accommodate up to 4.3% strain without plastic deformation. This is similar to Mg (bulk modulus = 45 GPa) which can accommodate about 5% strain at 100 °C before plastic deformation, depending on the crystal orientation of the film[47].

Mixtures or metastable alloys of immiscible metals have been attracting a lot of attention recently especially in catalysis because they exhibit interesting synergistic properties that are not obtainable in phase segregated mixtures[32, 48, 49]. Our results demonstrate that the addition of very minute amount of immiscible elements to a metal can lead to an unprecedented effects on the hydrogen sorption properties.

**Using YZr gradient films as a "hydrogen thermometer"**. An example of the practical relevance of this finding is demonstrated in Fig. 5. Here we show that the hydrogen-induced optical effects in YH$_x$ can be exploited to develop low-cost hydrogen detectors based on eye-readable color changes[23]. By using Y$_x$Zr$_{1-x}$ gradient films, we precisely tune the pressure at which the color change occurs. Figure 5 shows the colors observed in a Y$_{99}$Zr$_1$–Y$_{92}$Zr$_8$ gradient thin film as a function of hydrogen pressure. At ~0.05 mbar, the whole sample is in the YH$_{2.1}$ phase and exhibits a dark-grayish color everywhere. However, increasing the pressure gradually to 1000 mbar results in continuous coloration of the sample from left to right (low to high Zr concentrations) due to formation of YH$_3$ at a defined hydrogen pressure that increases with the Zr concentration (Supplementary Movies 1–3). The difference in the color of the fully hydrogenated region arises from interference effect, which depends on the film's thickness[23]. Thus, our discovery enables the realization of novel hydrogen sensors with a more than five orders of magnitude detection range at room temperature.

In summary, we have found a metal–hydrogen system with an exceptionally large and tunable thermodynamic destabilization induced by nanoscale physical constraints. In structurally coherent (Y–Zr)H$_x$ thin films, Zr nanoclusters induce an elastic strain on the yttrium, leading to destabilization of the trihydride (YH$_3$) phase. The equilibrium pressure for the formation of YH$_3$ is precisely and continuously tuned from $10^{-1}$ to $10^4$ mbar (five orders of magnitude) at room temperature by varying the Zr concentration from 0 to 13 at.%. Remarkably, the system is quite stable upon cycling and even exhibits a tunable equilibrium dehydrogenation pressure, an indication of true thermodynamic destabilization. Our research shows that the addition of certain

dopants, even if they are formally immiscible, can lead to unprecedented effects on the hydrogen sorption properties of metals. Of course, it remains to be confirmed that a similar effect can be attained in bulk materials as well. However, experiments performed by Asano et al. show that a similar coherency observed in MgTH$_x$ thin films can be reproduced in ball-milled bulk samples[40]. This opens a new avenue to rationally tailor the properties of metal–hydrogen systems to the desired applications. The findings might also be applicable to a wide range of reactions where intercalation into a coherent solid host lattice leads to a substantial volume change.

## Methods

**Sample preparation**. Thin films were prepared at room temperature in a multi-target ultrahigh-vacuum (UHV) DC/RF magnetron sputtering system with a base pressure of $10^{-9}$ mbar and a deposition pressure of 0.003 mbar Ar. The Y and Zr targets have a purity of 99.99% (4N) while the other targets (Gd, Pd, and Ti) have a purity of 99.9% (3N). The deposition rate of each metal is first determined by sputtering at a fixed power for 30 min and measuring the thickness of the deposited film using stylus profilometry (DEKTAK). For the gradient samples, the position-dependent deposition rate was measured along a $70 \times 5$ mm quartz substrate. Y–Zr gradient samples were prepared by sputtering Y and Zr from the opposite ends of a $70 \times 5 \times 0.5$ mm UV-grade quartz substrates at a predetermined rate, and without rotation, while the Y–Zr samples with uniform thickness and same Zr concentration where deposited on a $10 \times 10 \times 0.5$ mm UV-grade quartz substrate, which was rotated to ensure uniform deposition. A 5–10 nm Ti layer was deposited on the Y, Gd, Zr, (or Y–Zr, Gd–Zr) layer followed by a 10–30 nm Pd cap layer, which catalyzes hydrogen desorption and absorption, and prevents oxidation of the underlying layers. The Ti layer prevents alloying of Y and/or Zr with Pd.

**Structural characterization**. The phase composition and microstructure of the samples were characterized by in situ XRD measurements using a Bruker D8 Advance XRD system equipped with a Co X-ray source (0.178897 nm), and an Anton Paar in situ XRD reactor. The measurements were done at room temperature while the hydrogen pressure ranged from 0.01 to 10,000 mbar. Dehydrogenation of the hydrogenated sample at room temperature is achieved by flowing a 20% O$_2$/Ar gas mixture over the in situ cell, while at higher temperature (200–250 °C), this is achieved by gradually lowering the hydrogen pressure until it is below the plateau dehydrogenation pressure at that temperature.

**X-ray absorption spectroscopy (XAS)**. XANES and EXAFS measurements were done at the Dutch-Belgian Beam Line (DUBBLE) at the European Synchrotron Radiation Facility (ESRF), at the Yttrium and Zirconium K-edges (17,038 and 17,998 eV, respectively). The energy of the X-ray beam was tuned by a double-crystal monochromator operating in fixed-exit mode using a Si (111) crystal pair. Static measurements of the samples were performed in a closed-cycle He-cryostat (Oxford Instruments) at room temperature, and at 80 K to minimize the noise induced by the thermal Debye–Waller effect. The EXAFS spectra of the samples were collected at ambient temperature in fluorescence mode using a nine-element Ge detector (Ortec Inc.), whereas the spectra of Y, Zr, and Y$_2$O$_3$ reference foils were collected in transmission mode using Ar/He-filled ionization chambers. The threshold energy $E_k = 0$ was defined at 17,038 and 17,998 eV, respectively, for Y and Zr K-edge. Three scans per sample were measured at room temperature and at 77 K, energy-calibrated, averaged, and further analyzed using GNXAS[50, 51]. In this approach, the local atomic arrangement around the absorbing atom is decomposed into model atomic configurations containing $2…n$ atoms. The theoretical EXAFS signal $\chi(k)$ is given by the sum of the $n$-body contributions, $\gamma^3…\gamma^n$, which take into account all the possible single and multiple scattering (MS) paths between the $n$ atoms. The fitting of $\chi(k)$ to the experimental EXAFS signal allows to refine the relevant structural parameters of the different coordination shells; the suitability of the model is also evaluated by comparison of the experimental EXAFS FT with the FT of the calculated $\chi(k)$ function. The global fit parameters that were allowed to vary during the fitting procedure were the distance $R$(Å), Debye–Waller factor ($\sigma^2$) and the angles of the $\gamma^n$ contributions, which were defined according to the crystallographic structures used in the data analysis[52, 53]. There exist several polymorphic structures of zirconium hydride with similar atomic arrangement and Zr–Zr distances, but with slightly different lattice parameters[43, 54]. Accordingly, it is not trivial to establish unequivocally the crystallographic structure of the ZrH$_x$ nanoclusters, and the exact H concentration in the hydrogenated structure. Hence, we considered $\gamma_1^2$ as the Zr–H distance, $\gamma_2^2$ the Zr–Zr cubic enriched phase and $\gamma_3^2$ as the Zr–Zr distance of the tetragonal ZrH$_x$ phase.

**Hydrogen sorption measurement**. The thermodynamics and kinetics of hydrogen sorption in the thin films were measured using hydrogenography[24, 55] in transmission mode (reflection modes for the sensing experiments). This method is based on the fact that upon hydrogenation, the metal-to-insulator transition results in a substantial increase in the optical transmission of the films, with a maximum transmittance at the equilibrium plateau pressures as shown in Supplementary Figs. 1–4. This allows one to measure pressure-optical transmission isotherms (PTIs), which provide the same thermodynamic information as the typical pressure-composition-isotherms (PCI) measured for bulk metal hydrides. In this method, the amount of light transmitted or reflected by a thin film is measured as a function of hydrogen pressure at constant temperature. The normalized transmittance of the film $T/T_0$ is related to the hydrogen concentration $C_H$ and film thickness $d$ by Lambert–Beer law.

$$\ln(T/T_0) \propto C_H d \qquad (4)$$

Here $T$ is the optical transmission and $T_0$ is the initial transmittance of the film in the metallic state or, in dehydrogenated films, in the YH$_{1.9}$ phase before again a hydrogen pressure is applied[24, 55]. The samples were placed in a sample holder, and the sample holder placed in a closed cell with gas connections. The optical transmission or reflectance of the samples is measured using a white light source and a 3CCD camera. Transmission or reflection data of the samples with uniform layers are averaged across the whole sample, while for the $70 \times 5$ cm long gradient samples the data was only averaged in the direction perpendicular to the thickness gradient. The hydrogen pressure is gradually varied between 1 and $10^4$ mbar using a mixture of 0.1% H$_2$ in Ar, 4% H$_2$ in Ar, or pure H$_2$ at a flow rate of 20 ml min$^{-1}$. Typically, the time taken for an absorption isotherm is 8.5 h per order of magnitude in H$_2$ pressure, and in desorption 10 h per order of magnitude in H$_2$ pressure. The pressure steps are exponentially increasing or decreasing, so as to achieve a linear increase in the chemical potential of the gas. The logarithm of the change in optical reflection $\ln(R/R_0)$ or transmittance $\ln(T/T_0)$ is related to the hydrogen concentration in the Y film.

**Data availability**. All relevant data are available from the corresponding author on request.

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

## Acknowledgements

The authors acknowledge the financial support from the Netherlands Organisation for Scientific Research (NWO) for the beam time at the DUBBLE—Dutch-Belgian Beamline at ESRF Grenoble and Technology Foundation (STW) Netherland for a Valorization grant. Ing. Herman Schreuders is acknowledged for technical support. Dr Ruud Westerwaal, Dr Wim Haije, and Prof. Ronald Griessen are acknowledged for fruitful discussions.

## Author contributions

P.N. conceived, designed, and performed the experiments. W.B. and A.L. supported the EXAFS and XANES experiments. A.L. designed and performed the structural simulations. L.M. and B.D. made useful suggestions. P.N., A.L., and B.D. wrote and commented on the manuscript.

## Additional information

**Competing interests:** The authors declare no competing financial interests.

