## [Peer Review File · Nature Communications]

Reviewers' comments:

Reviewer #1 (Remarks to the Author):

The authors demonstrate a novel ability to tune the hydrogenation pressure of Y by several orders of magnitude by alloying with different amounts of Zr (1.5 at.% – 9 at.%) . This is a very interesting and useful property, which is shown by changing H₂ gas pressure and measuring the transmittance of the hydrides across a film with linearly varied Zr concentration. The authors propose that the mechanism responsible for this effect is lattice compression, as evidenced by XRD spectra showing a roughly 2-5% volume compression of the YH₂ phase due to the addition of Zr.

I see novelty and value in the demonstrated phenomena. The authors claim to have developed an ability to “rationally develop” alloying systems that tune the metal-hydride thermodynamics, and this relies on a proposed mechanism where Zr-nanoclusters pin grain boundaries and prevent plastic deformation. Unfortunately, this mechanism is largely based on conjecture and speculation, as no information on the location and size of the nanoclusters is provided. Furthermore, the notion that Zener pinning would effectively inhibit all available plastic deformation mechanisms is in itself speculated. This substantially detracts from the scope of the paper. Finally, the manuscript is not written in a clear manner, and thus interpretation of the main ideas requires significant effort.

Specific comments follow:

- 1) The evidence presented for Zr-nanoclusters is provided by XANES. What is the evidence for these Zr-nanoclusters successfully pinning grain boundaries? Electron microscopy images would benefit this claim tremendously. Moreover, just because grain boundary motion is impeded does not suggest that plastic deformation by other mechanisms is not preferred at such large lattice strains. What is the evidence that grain boundary pinning is the key ingredient to rational design of tunable metal-hydride systems?
- 2) An alternative explanation for the difference in hydrogenation pressure along the film, is a two-fold decrease in film thickness. For example, with smaller film thickness, the grain size of the film may be smaller, which may increase the barrier to the phase transition, or create local elastic fields near the grain boundary that have an effect.
- 3) The authors show the phase transition pressures from transmittance, but do not mention how much hydrogen is stored in the material, which is stated as one of the useful results of Zr-doping. In addition, it seems that Zr-doping increases the energy required to hydrogenate the material. How do the authors envision this ability to tunably increase the hydrogenation pressure to improve the hydrogen storage abilities of this material?
- 4) The manuscript is riddled with structure and grammar issues that significantly impede the understanding of the work. These issues include insufficient details and a confusing order of information as well as grammar, incomplete sentences, missing panel descriptions in figure captions, busy figures, etc.

Reviewer #3 (Remarks to the Author):

This is an interesting study that shows the evolution of H sorption pressure over a wide range by tailoring the composition and thickness of alloy films. The usage of an optical measurement tool to indirectly calculate H sorption is also an interesting technique. However the reviewer has the following concerns.

1. What is major impact of the current study? As pointed out by the reviewer, there is a great need for the discovery and design of advanced H storage materials that can store more than 5-7%

of H and allow rapidly hydrogenation/dehydrogenation at 100oC or less. Hence it is unclear why the authors keep on emphasizing that it is significant to tune H plateau pressure over several orders of magnitude. YZrH clearly does not have the weight capacity to achieve desirable H sorption.

2. The dehydrogenation temperature of these thin film remains high, ~ 200oC and in particular O₂ is necessary. The application of O₂ could potentially poison the films given the high reactivity of Y and Zr with O. Is the process reversible at all?

3. The technique adopted here has been used by the same group of authors in several other H sorption studies, and hence it is not a new technique (although it is still good to use it).

4. The justification of elastic clamping effect is not well supported. First, there is no in-depth calculation of film stress evolution. The authors have used some vague argument to indirectly infer that stress has been accumulated by examination of the peak position shift in XRD.

5. The argument of nanocluster pinning effect is not supported. Without any cross-section TEM micrograph and in-depth chemical analysis, it is difficult to decipher the evolution of microstructure and support the pinning effect by ZrH nanocluster along interfaces.

In summary, this will be a good story for a more specific journal. The novelty and impact demonstrated in the current study have not reached the level that can be categorized as a breakthrough in the field or significant/giant step forward.

Reviewer #4 (Remarks to the Author):

The manuscript by Ngene et al. presents an interesting and systematic study of a new strategy to destabilize metal hydrides by precipitation of nanoclusters (here Zr in Y matrix) at grain boundaries to prevent grain boundary motion and thus induce lattice strain, which in turn modifies the thermodynamics of the hydride. While engineering of metal hydrides by means of strain and, in particular, physically constraining thin films is not novel (see e.g. the related work by (at least) one of the authors: Baldi, A.; Gonzalez-Silveira, M.; Palmisano, V.; Dam, B.; Griessen, R. *Physical Review Letters* 2009, 102, (22), 226102. Molinari, A.; D'Amico, F.; Calizzi, M.; Zheng, Y.; Boelsma, C.; Mooij, L.; Lei, Y.; Hahn, H.; Dam, B.; Pasquini, L. *International Journal of Hydrogen Energy* 2016, 41, (23), 9841-9851.), the proposed concept of using grain boundary pinning by precipitation formation in combination with the obtained 5-orders of magnitude tunability of the (de)hydrogenation pressure is novel. Also, the demonstrated application as eye-readable hydrogen sensor with a very wide bandwidth is original and very innovative. For this reason, this work is of interest for the readers of *Nature Communications*. However, the authors have to satisfyingly address the following points:

General comments:

1. The employed hydrogenography method is indeed very powerful to probe a large number of different material compositions at the same time on one sample. However, since strain effects and lattice compressions are identified as the mediators of the observed effects I am lacking a discussion of the role of in-plane strain in the thin film. In other words, how can the authors be sure that each investigated data point that is ascribed to a certain stoichiometry of Y and Zr is not affected by the neighboring points in the same film? In their analysis each data point is treated as independent but this is hard to accept since all points are part of a continuous film.

2. What is the role of the support and strain induced at the interface towards the quartz? Could it be different for different stoichiometries of the film, e.g. due to different interfacial energies?

3. The authors state on page 5 that "Thus the addition of 12 at.% Zr leads to about five orders of magnitude increase in plateau pressure. This indicates an exponential relationship between the equilibrium hydrogen pressure and Zr concentration." As a consequence the strain energies in

these films must become very significant. Could the authors make a theoretical estimate of the strain built up in the film to obtain 5 orders of magnitude increase of plateau pressure? Is that a realistic number and at all commensurate with the fact that the authors seem to not observe any degradation and peeling of the films from the substrate even after numerous hydrogenation cycles?

4. Related to the question above, I wonder why strain is not relaxed by some alternative mechanism to plastic deformation via grain boundary movement, for example via nucleation and movement of dislocations within the crystallites. The authors should, at least speculatively, discuss this issue.

5. Why does the Y → YH_{1.9} transition pressure NOT depend on the Zr concentration?

6. On page 5 the authors state: "It has been shown that such gradient in sample thickness can result in different equilibrium pressures due to the differences in interfacial energy per unit mass¹⁸. However this contribution is not substantial in our sample (Fig. S1) hence the observed thermodynamic effect is unequivocally due to the presence of Zr." The thickness effect in shown in Figure S1 leads to basically 1 order of magnitude change of plateau pressure (from ca. 0.1 to ca. 2-3 mbar). I would not call that "not substantial" and it must thus be taken into account when discussing the data.

7. The main hypothesis put forward is that it is precipitation of Zr nanoclusters at grain boundaries that give rise to the massive tunability of the equilibrium pressure. It would most certainly strengthen the paper and argumentation very much if direct evidence of these nanoclusters and their accumulation at the grain boundaries would be presented, e.g. by high-resolution TEM.

8. For the sensor application it would certainly be very important to also briefly discuss the (de)hydrogenation kinetics of the present system as it relates to the response time of the hydrogen sensor. This is a key parameter to consider when assessing overall performance.

Specific detail comments:

1. First line page 5: 10-31 mbar should probably read 10-3 mbar.
2. Page 6, 4th line from the bottom: "inter-planner" should probably read "inter-planar".
3. Page 7, second last line – I would reconsider the use of the word "amazing".

Response to Reviewer's comments

We thank the reviewers for their time spent on reviewing our manuscript, and for their useful comments on our work. We have considered all their comments and have revised the paper accordingly. Point to point response to their comments are given below in *italics*. The manuscript has been thoroughly rewritten making it impossible to list all the changes. However, in the revised manuscript we highlighted in yellow those sections that are completely new.

Reviewers' comments:

Reviewer #1 (Remarks to the Author):

The authors demonstrate a novel ability to tune the hydrogenation pressure of Y by several orders of magnitude by alloying with different amounts of Zr (1.5 at.% – 9 at.%). This is a very interesting and useful property, which is shown by changing H₂ gas pressure and measuring the transmittance of the hydrides across a film with linearly varied Zr concentration. The authors propose that the mechanism responsible for this effect is lattice compression, as evidenced by XRD spectra showing a roughly 2-5% volume compression of the YH₂ phase due to the addition of Zr.

I see novelty and value in the demonstrated phenomena. The authors claim to have developed an ability to “rationally develop” alloying systems that tune the metal-hydride thermodynamics, and this relies on a proposed mechanism where Zr-nanoclusters pin grain boundaries and prevent plastic deformation. Unfortunately, this mechanism is largely based on conjecture and speculation, as no information on the location and size of the nanoclusters is provided. Furthermore, the notion that Zener pinning would effectively inhibit all available plastic deformation mechanisms is in itself speculated. This substantially detracts from the scope of the paper. Finally, the manuscript is not written in a clear manner, and thus interpretation of the main ideas requires significant effort.

Our response:

We thank the reviewer for agreeing with the novelty and value of our work. The referee's main concern is that Zener pinning, which we attributed as the origin of the observed change in plateau pressures, might not be the true cause of this effect. The other reviewers also share similar concern, hence we have conducted additional experiments and analysis to gain more insights into the nature of the ZrH_x clusters in the YZrH_x matrix. Based on the results from these additional experiments we did not observe any sign that suggests the segregation of the ZrH_x to the grain boundary of YH_x. Rather, it appears that the effect is due to lattice compression of YH_x induced

by ZrH_x nanoclusters in a structurally coherent metastable Y-Zr-H system. Surprisingly, the lattice compression remains intact during cycling. From a detailed analysis of the XANES results, we observed that the zirconium hydride clusters is highly distorted to maintain a coherent interface with the larger YH₃ and YH₂ after hydrogenation and dehydrogenation respectively. We have revised the paper accordingly based on this new understanding.

Specific comments follow:

1) The evidence presented for Zr-nanoclusters is provided by XANES. What is the evidence for these Zr-nanoclusters successfully pinning grain boundaries? Electron microscopy images would benefit this claim tremendously. Moreover, just because grain boundary motion is impeded does not suggest that plastic deformation by other mechanisms is not preferred at such large lattice strains. What is the evidence that grain boundary pinning is the key ingredient to rational design of tunable metal-hydride systems?

Our response

As stated above, we have now rewritten the paper based on the new results and understanding.

2) An alternative explanation for the difference in hydrogenation pressure along the film, is a two-fold decrease in film thickness. For example, with smaller film thickness, the grain size of the film may be smaller, which may increase the barrier to the phase transition, or create local elastic fields near the grain boundary that have an effect.

Our response

In supplementary Figure 1 we showed that an increase in the film thickness can indeed increase the barrier to phase transition and thereby an increase in the plateau hydrogenation pressure, as we previously reported in reference 14 (or reference 1 in supplementary information). However, this increase in plateau pressure is small when compared to the effect of Zr addition, and is not observed during dehydrogenation in contrast to the Zr doped samples. Furthermore, in supplementary Figure 4 we showed the PCIs of 5 different samples with same YZr thickness (60 nm) but different Zr concentrations, clearly, the plateau pressures of the samples depend strongly on the Zr content. This shows that the difference in plateau pressure is due to the addition of Zr.

3) The authors show the phase transition pressures from transmittance, but do not mention how much hydrogen is stored in the material, which is stated as one of the useful results of Zr-doping. In addition, it seems that Zr-doping increases the energy required to hydrogenate the material. How do the authors envision this ability to tunably increase the hydrogenation pressure to

improve the hydrogen storage abilities of this material?

Our response.

The method we used (hydrogenography) is able to provide information on the exact hydrogen content of thin films. The change in transmittance is related to the hydrogen content of the semiconducting hydride. Since the plateau width scales with the Y fraction, we are confident that the hydrogen storage capacity of yttrium is not affected by the doping. Thus, the storage capacity scales with the yttrium fraction. While the Zr also reversely absorbs hydrogen, this takes place at a different pressure and it depends on the application whether this can be put to use. The main focus of the work is to demonstrate that the thermodynamic of (de)hydrogenation can be tuned to a large extent using only a small amount of dopant.

On the comments about the increase on energy required to hydrogenate the Y due to Zr addition. This is indeed true: the YH_3 phase is destabilized, hence will hydrogenate and dehydrogenate at much higher pressures as compared to the pure YH_3 . It is the main objective of this work to show this. We explained this now in a more detail in the introduction section. As demonstrated in figure 5, this behavior is relevant for hydrogen sensing, and the same concept might be applicable to light-weight metal hydrides for reversible hydrogen storage.

4) The manuscript is riddled with structure and grammar issues that significantly impede the understanding of the work. These issues include insufficient details and a confusing order of information as well as grammar, incomplete sentences, missing panel descriptions in figure captions, busy figures, etc.

Our response.

We are sorry that we mistakenly submitted the wrong version of the manuscript which contained some grammatical and structural issues. We have largely rewritten the paper and have also corrected the grammatical and typing errors.

Reviewer #3 (Remarks to the Author):

This is an interesting study that shows the evolution of H sorption pressure over a wide range by tailoring the composition and thickness of alloy films. The usage of an optical measurement tool to indirectly calculate H sorption is also an interesting technique. However the reviewer has the following concerns.

1. What is major impact of the current study? As pointed out by the reviewer, there is a great

need for the discovery and design of advanced H storage materials that can store more than 5-7% of H and allow rapidly hydrogenation/dehydrogenation at 100oC or less. Hence it is unclear why the authors keep on emphasizing that it is significant to tune H plateau pressure over several orders of magnitude. YZrH clearly does not have the weight capacity to achieve desirable H sorption.

Our response

We agree with the referee that the hydrogen content of YH₃ is less than the 5-7 wt% required for hydrogen storage application. However, as we pointed out in the introduction section, there have been several efforts to rationally alter the enthalpy of hydrogen sorption in metals, and thereby the equilibrium (de)hydrogenation pressure and temperature of metal hydrides. This is crucial for application. Our YZr system is a model system that shows for the first time that a large thermodynamic destabilization can be achieved in metal hydrides by a lattice compression which is induced by small amounts of dopant materials. We expect that similar effects can be achieved in light-weight metal hydrides, such as MgH₂ and LiH when the appropriate dopant material(s) is used. We demonstrate here the relevance of our finding in hydrogen sensing application.

2. The dehydrogenation temperature of these thin film remains high, ~ 200oC and in particular O₂ is necessary. The application of O₂ could potentially poison the films given the high reactivity of Y and Zr with O. Is the process reversible at all?

Our Response

In figure 1d we show that the equilibrium dehydrogenation pressure (at 200 °C) of our samples increases as the Zr concentration increases. This means that the equilibrium dehydrogenation temperature decreases with increasing Zr concentration. We used (200 °C - 220°C) for the desorption experiments in order to cover the whole temperature range for the different Zr concentrations of the gradient sample.

For the remark on the use of oxygen for desorption, it appears that we did not make it very explicit in the paper that all the dehydrogenation experiments were conducted under hydrogen atmosphere, except for the sensing experiments where 20 % oxygen/Ar gas mixture was used. This was done to understand the dehydrogenation kinetics of the sensors when used under normal atmospheric conditions. We have shown in our previous work (ref 23) that oxygen is not detrimental to the sensors because the Pd top layer protects the YZr sensing layer from oxidation, as also explained in the manuscript. Finally, the process is very reversible, irrespective of the gas used for dehydrogenation (H₂ or O₂)

3. The technique adopted here has been used by the same group of authors in several other H sorption studies, and hence it is not a new technique (although it is still good to use it).

Our Response

The reviewer is indeed right that we have used this technique for several other related studies. The focus of the present work is not on the technique but on the fact that we used it to discover new metal-hydrogen systems that exhibit unique thermodynamic properties.

4. The justification of elastic clamping effect is not well supported. First, there is no in-depth calculation of film stress evolution. The authors have used some vague argument to indirectly infer that stress has been accumulated by examination of the peak position shift in XRD.

Our response:

Although XRD is not the most ideal technique to quantify elastic clamping effects, it is very reliable in determining the lattice spacing (parameters) of crystalline materials. We based the elastic clamping effect on three important observations. First, the lattice parameters of Y decreases with increasing Zr content, suggesting that the lattice is compressed due to the smaller atomic size of Zr. Secondly, we observed that this lattice compression remains intact after hydrogenation and dehydrogenation, showing that the compression is elastic. Third, for any given Zr composition, the strain energy due to the lattice compression is proportional to the increase in equilibrium hydrogenation pressure, as would be expected based on equation 3 in the main paper. Additional experiments suggest that this effect is due to the formation of coherent YZr phase due to the small sizes of the ZrH_x clusters. We have added this additional information in the revised manuscript.

5. The argument of nanocluster pinning effect is not supported. Without any cross-section TEM micrograph and in-depth chemical analysis, it is difficult to decipher the evolution of microstructure and support the pinning effect by ZrH nanocluster along interfaces.

Our response:

This point has been discussed above (first response to reviewer # 3). As mentioned above, additional experimental results shows no evidence for our suggestion that the ZrH_x clusters segregate to the grain boundary. Instead, we found strong indications that they form a coherent interface with the YH_x matrix, resulting in compression (strain) of the YH_x . The interface seem to be stable during cycling, explaining why the lattice compression remained intact during hydrogenation and dehydrogenation. We have explained this in the revised manuscript.

In summary, this will be a good story for a more specific journal. The novelty and impact demonstrated in the current study have not reached the level that can be categorized as a breakthrough in the field or significant/giant step forward.

Our response

Our work shows for the first time that the thermodynamics, hence (de)dehydrogenation temperature and pressures of metal hydrides can be significantly altered by lattice compression induced by adding small quantities of dopant material which is insoluble in the metal hydride. We believe that this is a novel and very useful work which can lead to the development of new light-weight metal hydrides for various applications. The comments of the other reviewers also attest to the value and impact of our work. Therefore we are confident that this work is now suitable for publication in Nature communication.

Reviewer #4 (Remarks to the Author):

The manuscript by Ngene et al. presents an interesting and systematic study of a new strategy to destabilize metal hydrides by precipitation of nanoclusters (here Zr in Y matrix) at grain boundaries to prevent grain boundary motion and thus induce lattice strain, which in turn modifies the thermodynamics of the hydride. While engineering of metal hydrides by means of strain and, in particular, physically constraining thin films is not novel (see e.g. the related work by (at least) one of the authors: Baldi, A.; Gonzalez-Silveira, M.; Palmisano, V.; Dam, B.; Griessen, R. *Physical Review Letters* 2009, 102, (22), 226102. Molinari, A.; D'Amico, F.; Calizzi, M.; Zheng, Y.; Boelsma, C.; Mooij, L.; Lei, Y.; Hahn, H.; Dam, B.; Pasquini, L. *International Journal of Hydrogen Energy* 2016, 41, (23), 9841-9851.), the proposed concept of using grain boundary pinning by precipitation formation in combination with the obtained 5-orders of magnitude tunability of the (de)hydrogenation pressure is novel. Also, the demonstrated application as eye-readable hydrogen sensor with a very wide bandwidth is original and very innovative. For this reason, this work is of interest for the readers of Nature Communications. However, the authors have to satisfyingly address the following points:

General comments:

1. The employed hydrogenography method is indeed very powerful to probe a large number of different material compositions at the same time on one sample. However, since strain effects and lattice compressions are identified as the mediators of the observed effects I am lacking a discussion of the role of in-plane strain in the thin film. In other words, how can the authors be sure that each investigated data point that is ascribed to a certain stoichiometry of Y and Zr is not affected by the neighboring points in the same film? In their analysis each data point is treated as independent but this is hard to accept since all points are part of a continuous film.

Our response:

This is certainly an interesting point. For this reason we also measured the PTIs of 5 different samples (deposited on 1 cm x 1cm quartz substrate) with same YZr thickness (60 nm) but different Zr concentrations. As shown in supplementary Figure 4, the plateau pressures of the samples depend strongly on the Zr content as observed for the gradient films. This shows the difference in plateau pressure is indeed due to the addition of Zr, and that the in-plane strain in the thin film did have a large influence on the observed thermodynamic effects.

2. What is the role of the support and strain induced at the interface towards the quartz? Could it be different for different stoichiometries of the film, e.g. due to different interfacial energies?

Our response:

This question is partly answered by the explanation given for point 1 above. In addition, we showed in supplementary Figure 1 that an increase in the film thickness can lead to an increase in the plateau hydrogenation pressure due to different interfacial energy, as we previously reported in reference 14 (or reference 1 in supplementary information). However, this increase in plateau pressure is negligible when compared to the effect of Zr addition, and is not observed during dehydrogenation, in contrast to the Zr doped samples. Thus the effect of Zr is much stronger.

3. The authors state on page 5 that “Thus the addition of 12 at.% Zr leads to about five orders of magnitude increase in plateau pressure. This indicates an exponential relationship between the equilibrium hydrogen pressure and Zr concentration.” As a consequence the strain energies in these films must become very significant. Could the authors make a theoretical estimate of the strain built up in the film to obtain 5 orders of magnitude increase of plateau pressure? Is that a realistic number and at all commensurate with the fact that the authors seem to not observe any degradation and peeling of the films from the substrate even after numerous hydrogenation cycles?

Our response:

We deduce stresses increasing up to 6 GPa as a result of the Zr doping. This is larger than the 1 GPa that we deduced for the yield stress of pure YH_x films (J. Mater. Chem., 2012, 22, 24453–24462). The additional pinning to plastic behavior could be due to the Zr-based impurities, however the proof for this is outside the scope of the present paper. We added a discussion to the paper.

4. Related to the question above, I wonder why strain is not relaxed by some alternative mechanism to plastic deformation via grain boundary movement, for example via nucleation and

movement of dislocations within the crystallites. The authors should, at least speculatively, discuss this issue.

Our response:

Clearly, the fact that the Y-lattice compression is reversible indicates that we are dealing with elastic effects. It is remarkable that this compression is maintained upon repeated cycling. This cycling involves volume changes larger than 10% and definitely leads to plastic deformation. As we discuss in the paper, the change in the desorption pressure is much smaller than expected. How the two effects (elastic compression by Zr, and plastic deformation during the phase transition) interact requires a more single-crystal type of sample, which is outside our present capabilities.

5. Why does the Y → YH_{1.9} transition pressure NOT depend on the Zr concentration?

Our response:

As we explained on page 4 the equilibrium pressure for Y → YH_{1.9} is $\sim 10^{-31}$ mbar but we observe this transition around 5×10^{-3} mbar due to severe kinetic limitations. The increase in the thermodynamics of the dihydride formation by a couple of orders in pressure is unlikely to have any effect on the kinetics.

6. On page 5 the authors state: “It has been shown that such gradient in sample thickness can result in different equilibrium pressures due to the differences in interfacial energy per unit mass¹⁸. However this contribution is not substantial in our sample (Fig. S1) hence the observed thermodynamic effect is unequivocally due to the presence of Zr.” The thickness effect in shown in Figure S1 leads to basically 1 order of magnitude change of plateau pressure (from ca. 0.1 to ca. 2-3 mbar). I would not call that “not substantial” and it must thus be taken into account when discussing the data.

Our response:

What we argue here is that the change in plateau pressure of YH₃ due to the thickness effect is not substantial when compared to the effect of Zr concentration as demonstrated in supplementary figures 1 and 4 where we show the plateau pressure of samples with same thickness but varying Zr concentrations. This is also true especially when we consider the dehydrogenation behavior of this sample. We have changed the sentence to reflect the fact that the comparison is between Zr doped and non-doped Y samples.

7. The main hypothesis put forward is that it is precipitation of Zr nanoclusters at grain boundaries that give rise to the massive tunability of the equilibrium pressure. It would most

certainly strengthen the paper and argumentation very much if direct evidence of these nanoclusters and their accumulation at the grain boundaries would be presented, e.g. by high-resolution TEM.

Our response

We have attempted to obtain more insights into the structure of the Zr by combining high resolution TEM and FIB-SEM-EDX analysis. We found so far no evidence for the proposed grain boundary pinning which lead to a revision of our argumentation

8. For the sensor application it would certainly be very important to also briefly discuss the (de)hydrogenation kinetics of the present system as it relates to the response time of the hydrogen sensor. This is a key parameter to consider when assessing overall performance.

Our response

The response kinetics for an Y-H based sensor in different environment has been described in details in reference 23. The addition of Zr did not change the response kinetics. However, the present work is focused on the concept of a “hydrogen thermometer” using $YZrH_x$ based thin films. Since the different Zr concentration requires different hydrogen pressures to change color, we only mentioned (in supplementary video 1 and 2) the time required for the gradient sample to change color as the pressure is increased. At each concentration, the response is as fast as for the YH_x sensor.

Specific detail comments:

1. First line page 5: 10⁻³¹ mbar should probably read 10⁻³ mbar.
2. Page 6, 4th line from the bottom: “inter-planner” should probably read “inter-planar”.
3. Page 7, second last line – I would reconsider the use of the word “amazing”.

Our response

We have corrected these errors in the revised manuscript.

Reviewers' Comments:

Reviewer #3:

Remarks to the Author:

The authors have reanalyzed their XANES data and stated that for the first time that a large thermodynamic destabilization can be achieved in metal hydrides by a lattice compression which is induced by small amounts of dopant materials.

Such a statement is certainly incorrect. The influence of lattice compression on destabilization of other hydrides, such as MgH_2 , has been demonstrated a few years back.

Furthermore the conclusion on development of compressive stress is largely based on curve fitting. Are the authors sure there is only one unique solution to the fitting? Without showing direct evidence of the so-called coherent ZrH nanocluster in YH, their statement is weakly supported.

In view of these questions and concerns, the reviewer cannot recommend the paper for publication.

Reviewer #4:

Remarks to the Author:

The authors have satisfyingly addressed my concerns from the first round of revision and I can now, in principle, recommend publication. However, the text is still riddled with missprints, strange formulations and generally sometimes unsatisfying English. I thus urge the authors to carefully revise the text as such (the science is OK) and maybe make use of a language editor.

I have also been asked by the editor to take position to the response of the authors to the concerns of Reviewer #1, which I will do in this separate section.

To my understanding the most critical concern raised by Reviewer # 1 was the questioning of Zener pinning being the mechanism by which the vast tuning of the equilibrium pressures is facilitated. The authors have indeed addressed this concern in detail by now explaining the observation based on a different mechanism which is well-supported by the provided experimental results. Hence this comment is adequately addressed.

The same goes for comment #2 which is well taken care of by the data provided.

When it comes to comment #3, in principle I can accept the response of the authors with the small exception that it still is not clear to me how the authors, as stated in the reply, are able to provide the exact hydrogen content in the film from their hydrogenography measurements.

As I also indicated in my response, I agree with Reviewer #1 that the text was (and actually still is) riddled with language issues and strange formulations that should be addressed.

Generally, also on the basis of the response to Reviewer #1, I can recommend the paper for publication provided the language issues are addressed.

Reviewer #5:

Remarks to the Author:

The manuscript by Peter Ngene et al. presents a very thorough investigation on a thin-film metal-hydrogen system. The measurements and characterization is detailed and excellent and the discussion is sound. However, the author express or give the impression that this is a general

property of this metal-hydrogen system, but their investigations are performed on thin-film systems. The film is less than 100nm and the results maybe strongly influenced by this limited size in two dimension. The samples are prepared by sputtering and, therefore, are not in thermal equilibrium. I don't expect Zr nanoclusters in a bulk material prepared by conventional metallurgical processing and, therefore, their results are only characteristic and representative for non-equilibrium thin-film systems. Nevertheless, their results are very important, e.g., for hydrogen sensors. In the last sentence of the abstract they mention this, however, somehow this sentence contains some grammatical error.

In general, I highly recommend to publish the manuscript after minor revision, i.e. the authors should emphasize that this is a thin-film system and a generalization to metal-hydrogen systems in thermodynamic equilibrium may not be possible.

Response to Reviewer's comments

We thank the reviewers for the time spent on reviewing our manuscript and for their useful comments. Below is the point by point response to their comments.

Reviewer #3 (Remarks to the Author):

The authors have reanalyzed their XANES data and stated that for the first time that a large thermodynamic destabilization can be achieved in metal hydrides by a lattice compression which is induced by small amounts of dopant materials.

Such a statement is certainly incorrect. The influence of lattice compression on destabilization of other hydrides, such as MgH₂, has been demonstrated a few years back.

Our Response:

Without a proper reference it is hard to reflect on this statement. Indeed, many computational studies have been done in this direction. Experimental studies usually only show the effect on the desorption temperature, which is only an indirect indication of destabilization, due to the kinetic component in such an experiment. We are not aware of any studies relating a lattice compression to such a large change in the equilibrium pressure as we observe.

Reviewer #3

Furthermore the conclusion on development of compressive stress is largely based on curve fitting. Are the authors sure there is only one unique solution to the fitting? Without showing direct evidence of the so-called coherent ZrH nanocluster in YH, their statement is weakly supported.

In view of these questions and concerns, the reviewer cannot recommend the paper for publication.

Our Response:

The development of compressive stress is based on our XRD data which clearly shows a compression of the yttrium lattice. No fitting is involved. The nature of the ZrH cluster inducing this compression is indeed the result of a fitting of the XAS. This was done using the SCF procedure available in FDMNES package and then compared with the experimental data. Using a well-established structural model combined with extended calculations, we obtained very reliable results (good fit to the experimental data) for the samples in the as-prepared and dehydrogenated states. In addition, the EXAFS results shown in the supplementary information for the Y phases, are well in line with the XRD results. However, in the case of Zr, it is only by adopting a highly distorted ZrH_x structure (similar to the hexagonal YH₃ environment) that the spectra of ZrH_x in the hydrogenated state are reproduced. No other fittings could reliably reproduce the measured spectra; hence we conclude that the Zr indeed adopts the distorted configuration to fit with the hexagonal YH₃ environment, in line with the coherent Y-Zr phase suggested by the XRD results.

Note that if the referee implies that TEM would constitute a more direct evidence: also in that case one would need a fitting procedure to establish the nature of such a structure.

Reviewer #4 (Remarks to the Author):

The authors have satisfyingly addressed my concerns from the first round of revision and I can now, in principle, recommend publication. However, the text is still riddled with missprints, strange formulations and generally sometimes unsatisfying English. I thus urge the authors to carefully revise the text as such (the science is OK) and maybe make use of a language editor.

I have also been asked by the editor to take position to the response of the authors to the concerns of Reviewer #1, which I will do in this separate section.

To my understanding the most critical concern raised by Reviewer #1 was the questioning of Zener pinning being the mechanism by which the vast tuning of the equilibrium pressures is facilitated. The authors have indeed addressed this concern in detail by now explaining the observation based on a different mechanism which is well-supported by the provided experimental results. Hence this comment is adequately addressed.

The same goes for comment #2 which is well taken care of by the data provided.

When it comes to comment #3, in principle I can accept the response of the authors with the small exception that it still is not clear to me how the authors, as stated in the reply, are able to provide the exact hydrogen content in the film from their hydrogenography measurements.

As I also indicated in my response, I agree with Reviewer #1 that the text was (and actually still is) riddled with language issues and strange formulations that should be addressed.

Generally, also on the basis of the response to Reviewer #1, I can recommend the paper for publication provided the language issues are addressed.

Our Response:

We carefully revised the text as is evidenced from the many small changes in ‘track revisions’.

Referee #5

However, the author express or give the impression that this is a general property of this metal-hydrogen system, but their investigations are performed on thin-film systems. The film is less than 100nm and the results maybe strongly influenced by this limited size in two dimension. The samples are prepared by sputtering and, therefore, are not in thermal equilibrium. I don't expect Zr nanoclusters in a bulk material prepared by conventional metallurgical processing and, therefore, their results are only characteristic and representative for non-equilibrium thin-film systems.

Our Response:

We acknowledge the thin film nature of our results, but added a sentence in the last paragraph pointing to the reported similarity between thin films and ball milled bulk samples.

Nevertheless, their results are very important, e.g., for hydrogen sensors. In the last sentence of the abstract they mention this, however, somehow this sentence contains some grammatical error.

Our Response:

We improved the last sentence of the abstract

In general, I highly recommend to publish the manuscript after minor revision, i.e. the authors should emphasize that this is a thin-film system and a generalization to metal-hydrogen systems in

thermodynamic equilibrium may not be possible.

Our Response:

See above for the thin film/bulk discussion

In addition, the following changes have been applied to the manuscript according to you're the suggestions of the editor:

- Main text is rewritten with track changes. The “Results” section is divided by subsections. There is no separate Discussion section
- The information about the movie has been supplied as a separate document, as requested.
- Information about competing interests (financial) and authors contribution has been added.